# Repeated Rounds of Gonadotropin Stimulation Induce Imbalance in the Antioxidant Machinery and Activation of Pro-Survival Proteins in Mouse Oviducts

**DOI:** 10.3390/ijms24119294

**Published:** 2023-05-26

**Authors:** Valentina Di Nisio, Sevastiani Antonouli, Sabrina Colafarina, Osvaldo Zarivi, Gianna Rossi, Sandra Cecconi, Anna Maria Giuseppina Poma

**Affiliations:** 1Department of Gynecology and Reproductive Medicine, Karolinska University Hospital, SE-14186 Stockholm, Sweden; valentina.di.nisio@ki.se; 2Division of Obstetrics and Gynecology, Department of Clinical Science, Intervention and Technology, Karolinska Institutet, SE-14186 Stockholm, Sweden; 3Department of Clinical Chemistry, Faculty of Medicine, University of Ioannina, PC-45110 Ioannina, Greece; arella_935@hotmail.com; 4Department of Life, Health and Environmental Sciences, University of L’Aquila, 67100 L’Aquila, Italy; sabrina.colafarina@univaq.it (S.C.); osvaldo.zarivi@univaq.it (O.Z.); gianna.rossi@univaq.it (G.R.); annamariagiuseppina.poma@univaq.it (A.M.G.P.)

**Keywords:** oviducts, repeated ovarian stimulation, mtDNA fragmentation, antioxidant enzymes, pro-survival signaling

## Abstract

Controlled ovarian stimulation (COS) through gonadotropin administration has become a common procedure in assisted reproductive technologies. COS’s drawback is the formation of an unbalanced hormonal and molecular environment that could alter several cellular mechanisms. On this basis, we detected the presence of mitochondrial DNA (mtDNA) fragmentation, antioxidant enzymes (catalase; superoxide dismutases 1 and 2, SOD-1 and -2; glutathione peroxidase 1, GPx1) and apoptotic (Bcl-2-associated X protein, Bax; cleaved caspases 3 and 7; phosphorylated (p)-heat shock protein 27, p-HSP27) and cell-cycle-related proteins (p-p38 mitogen-activated protein kinase, p-p38 MAPK; p-MAPK activated protein kinase 2, p-MAPKAPK2; p-stress-activated protein kinase/Jun amino-terminal kinase, p-SAPK/JNK; p-c-Jun) in the oviducts of unstimulated (Ctr) and repeatedly hyperstimulated (eight rounds, 8R) mice. While all the antioxidant enzymes were overexpressed after 8R of stimulation, mtDNA fragmentation decreased in the 8R group, denoting a present yet controlled imbalance in the antioxidant machinery. Apoptotic proteins were not overexpressed, except for a sharp increase in the inflammatory-related cleaved caspase 7, accompanied by a significant decrease in p-HSP27 content. On the other hand, the number of proteins involved in pro-survival mechanisms, such as p-p38 MAPK, p-SAPK/JNK and p-c-Jun, increased almost 50% in the 8R group. Altogether, the present results demonstrate that repeated stimulations cause the activation of the antioxidant machinery in mouse oviducts; however, this is not sufficient to induce apoptosis, and is efficiently counterbalanced by activation of pro-survival proteins.

## 1. Introduction

In recent years, the infertility rate in humans has increased significantly and the estimated percentage of reproductive-aged couples having fertility problems is about 12% [1]. Complex in vitro technologies have been developed for ameliorating female fertility potential [2]. From a clinical routine perspective, the most used technologies are generally based on the administration of fertility drugs, such as pituitary gonadotropins, to obtain a high number of oocytes. More recently, oocyte donation programs have been frequently offered to meet the demand of women with hereditable diseases, or cancer, or those of an advanced reproductive age [3]. To support these programs, young healthy donors often undergo multiple cycles of ovarian stimulation to bypass the limitation of oocyte yield in such unfortunate patients [4].

There is great concern about the risk for these stimulated women of developing the inflammatory state of ovarian hyperstimulation syndrome [5] or reproductive cancers, i.e., breast, ovarian and endometrial cancer [6,7]. As a matter of fact, ovarian cancer (OC) is a hormone-related cancer, as oral contraceptive use and parity are well-established protective factors [8]. Epidemiological data show that the incidence of OC increases when gonadotropins are elevated, as occurs in menopause transition [9]. An increased number of ovulations [10] and high gonadotropin levels, as found in nulliparous women, are theorized to be factors of OC development [11,12].

To date, the use of repetitive stimulatory protocols, especially those based on gonadotropins, has not been unequivocally correlated with the onset of OC. An updated Cochrane review [13] reported the absence of an increased risk for invasive OC after fertility drug treatments; however, in subfertile women, the occurrence of borderline ovarian tumors cannot be excluded, which presents a consistent risk for transformation in malignant OC [14]. Accordingly, in their final comments, the authors highlighted the need for other studies to either overrule or confirm this threat, as there is insufficient information on the dose and/or type of stimulation and a lack of a congruous number of studies conducted to prove if and to which extent fertility drugs are involved in OC outbreak.

Findings from several studies provide evidence that the most common type of OC, high-grade serous ovarian carcinoma (HGSOC), may originate in the oviducts. In fact, the rapid spread of serous tubal intraepithelial carcinomas (STICs) to the ovarian surface is followed by the development of HGSOC [15,16,17]. This theory is confirmed by the finding that tubal ligation and salpingectomy strongly reduce the risk of developing this deadly cancer [18] and by the high expression of many oviductal genes in these cancers [19,20]. Recently, a dualistic origin of HGSOC has been proposed based on comparative analysis of tumors derived from the oviductal epithelium and ovarian superficial cells, revealing differences in latency, metastatic behavior, and molecular characteristics [21].

In view of this, it is of interest to elucidate if and how repeated stimulatory cycles with fertility drugs target the oviducts and induce molecular alterations. Our previous studies showed that the stimulation of female mice with gonadotropins for eight rounds (8R) leads to overexpression of selected key proteins involved in the stimulation of cell cycle progression and survival in the oviducts, but not in the ovaries [22,23]. Ultrastructural analysis of the hyperstimulated oviductal epithelium showed altered morphological aspects of tubal ciliated cells, such as the presence of swollen mitochondria and the absence of cilia [24]. Considering this evidence and previous studies detecting the activation of oxidative stress markers in mouse cumulus cells [25] and in the serum of women undergoing assisted reproductive technologies [26], we analyzed here how repeated stimulatory cycles with gonadotropins could target the oviducts to initiate oxidative stress, causing the activation of serial processes that may trigger the initiation of cancer-related alterations in the future. To this end, we investigated the extent of mitochondrial (mt)DNA damage (mtDNA fragmentation, DNA polymerase γ), the content of antioxidant enzymes (catalase; superoxide dismutases 1 and 2, SOD-1 and -2; glutathione peroxidase 1, GPx1) and the activation of pro-apoptotic proteins (Bcl-2-associated X protein, Bax; cleaved caspases 3 and 7; phosphorylated (p)-heat shock protein 27, p-HSP27) as well as that of pro-survival proteins (p-p38 mitogen-activated protein kinase, p-p38 MAPK; p-MAPK activated protein kinase 2, p-MAPKAPK2; p-stress-activated protein kinase/Jun amino-terminal kinase, p-SAPK/JNK; p-c-Jun).

## 2. Results

### 2.1. Protein Content of Oxidative Stress Enzymes in Unstimulated and Hyperstimulated Mouse Oviducts

Experiments were performed to evaluate if the protein content of the main enzymes protecting cells from oxidative damage could be differentially modulated after 8R as compared with Ctr. As shown in Figure 1A,B, repeated stimulations induced a significant increase in catalase (+31%), GPx1 (+40%), SOD-1 (+33%) and SOD-2 (+32%) contents over Ctr (*p <* 0.05).

### 2.2. Mitochondrial Damage in Unstimulated and Hyperstimulated Mouse Oviducts

Since the results from ultrastructural analysis revealed that 8R of gonadotropin stimulations modified the morphological characteristics of mitochondria [24], in these experiments the occurrence of mtDNA damage was assessed by quantifying DNA polymerase γ content and 8.2 kb mtDNA fragments in oviducts collected from unstimulated (Ctr) and hyperstimulated (8R) mice. The results showed that the quantity of 8.2 Kb mtDNA fragments decreased slightly but significantly in 8R mice (Figure 2A; −19% vs. Ctr; *p* < 0.05), while DNA polymerase γ protein content was unaffected by hormone administration (Figure 2B,C; *p* > 0.05).

### 2.3. Content of Apoptotic Proteins in Unstimulated and Hyperstimulated Mouse Oviducts

In these experiments, it was assessed whether 8R could stimulate apoptotic processes. To this end, Bax and cleaved caspases 3 and 7 contents were determined, together with that of p-HSP27 because of its ability to interact with components of the apoptotic signaling pathway involved in caspase activation [27]. The results in Figure 3 evidenced that while Bax and cleaved caspase 3 contents were comparable between treated and Ctr groups (Figure 3B; +12% and +8%, respectively; *p >* 0.05), that of cleaved caspase 7 and p-HSP27 underwent a significant increase and decrease, respectively. Indeed, a 2.5-fold accumulation of cleaved caspase 7 (*p <* 0.05) occurred simultaneously with a drastic reduction in HSP27 content as compared with Ctr (−50%; *p <* 0.05).

### 2.4. Content of Cell-Cycle-Related Proteins in Unstimulated and Hyperstimulated Mouse Oviducts

The analysis of cell-cycle-related proteins revealed that the MAPK pathway was activated after 8R of stimulation. The oviducts after 8R of stimulation showed higher levels of p-MAPKAPK2, p-p38 MAPK, p-SAP/JNK and p-c-JUN in comparison with Ctr (Figure 4). In particular, p-MAPKAPK2 content increased about 24% (*p <* 0.05), while that of p-p38 MAPK, p-SAPK/JNK and p-c-JUN almost doubled (+42%, +42% and +49%, respectively; *p <* 0.05).

## 3. Discussion

Ovulation is an inflammatory reaction occurring in mature follicles, as it can be inhibited by nonsteroidal anti-inflammatory drugs [28]. At low levels, reactive oxygen species (ROS) regulate the physiological process of ovulation by promoting follicle wall breakdown; conversely, at high levels, they affect follicle development by reducing steroidogenesis and inducing atresia [29]. Furthermore, during aging, an increase in the intracellular levels of ROS and a concomitant decrease in major antioxidant enzyme synthesis have been described in the female reproductive system, creating an imbalance in the follicular compartment and affecting the quality and competence of oocytes. In fact, the high ROS levels detected in the follicular fluids and granulosa cells of aged women undergoing assisted reproductive technology procedures reduce both their fertilization rate and final pregnancy outcome [30,31]. These data highlight the importance of the fine balance in ROS follicular levels, suggesting that an impairment in the antioxidant machinery could interfere with oocyte quality and both fertilization and pregnancy processes.

It has been reported that repeated administration of gonadotropins causes excessive exposure to oxidative stress of the whole mouse ovary. In fact, although the ovarian protein level of catalase was unaffected by repeated ovulation, that of SOD-1 and SOD-2 decreased, with SOD-2 being restored to normal levels after the administration of L-carnitine [32]. Moreover, a dramatic reduction in oocyte quality was recorded due to oocyte mitochondria and spindle organization impairment by repeated ovulation [22,23,32].

The effects of multiple hormonal stimulation cycles on the oxidative stress of the oviducts are of interest, since these organs are not only exposed to high levels of ROS present in the follicular fluid but are also the place of fertilization in early embryonic development and of STICs’ etiology. In mouse oviducts hyperstimulated for eight repetitive rounds of gonadotropins (8R), we found that a pool of key cell cycle proteins (p-AKT, p-GSK3B, p-p53, cyclin D1 and OCT3/4) was overexpressed [22,23], thereby raising interrogatives on other mechanisms potentially dysregulated by this procedure. Here, we found that the antioxidant enzymes SOD-1 and -2 and catalase and GPx1 were overexpressed in the 8R mouse oviducts. These enzymes are physiologically present in the oviductal fluid of mice, cows and humans to protect embryos from oxidative damage [33]. GPx1 has also been detected in human ovaries, where its polymorphism has been associated with polycystic ovary syndrome [34], and in the ovaries of aged mice, where it is usually downregulated unless antioxidant supplementation is employed [35]. From a broader perspective, all antioxidant enzymes appear to be modulated during ovulation, and even more so during controlled ovarian hyperstimulation [26]. This mechanism involves the hormonal pulse linked to estrogens and to the function of estrogen receptors in oviducts after administration of gonadotropins, which mobilizes the antioxidant defense system, thus maintaining the redox homeostasis in the female reproductive tract [26,36].

The altered mitochondria ultrastructure in oviductal cells [24] prompted us to investigate the occurrence of mtDNA damage by quantifying firstly the 8.2 kb products and then the content of DNA polymerase γ, which is involved in mtDNA replication and repair as well as in the maintenance of the correct mitochondrial numbers [37]. In mouse cumulus cells, Xie and collaborators found that repeated superovulation adversely affects mitochondrial functions by changing DNA polymerase γ A gene expression [25]. Our results clearly indicate the absence of extensive mtDNA damage because the slight, although significant, reduction in its integrity was not accompanied by changes in DNA polymerase γ protein content as compared with the control. This suggests that the mtDNA copy number is not dramatically affected and that eventual damage can be repaired [38]. Moreover, the absence of extensive stimulation-dependent nuclear DNA damage is supported by our previous finding of the low content of p-Chk1 and p-H2A.X [22].

A variety of stress conditions, including temperature shock, oxidative stress, inflammation, and tumor development can affect the response of molecular chaperones known as heat shock proteins (HSPs). Mammalian small HSPs, such as HSP90 and HSP27, function both as antioxidants and anti-apoptotic agents [39]. The main regulation of HSP27 is orchestrated via the phosphorylation by MAPKAP kinase 2/3 and the p38 MAPK pathway [39]. Moreover, the phosphorylated form of HSP27 has been found to modulate estrogen signaling by interacting with estrogen receptor β [40]. However, in mouse oviductal epithelia, 8R of hyperstimulation induces a significant decrease in p-HSP27 protein content. Notably, literature data proved the localization of both HSP90 and 27 in ciliated cells of diverse organs, including the nasal, endometrium and oviductal epithelium [41,42,43]. Therefore, our finding is supported by ultrastructural data showing a drastic decrease in ciliated cells following 8R [24]. As a response to cellular stress, p-HSP27 promotes cell protection and prevents apoptosis by interacting with AKT, thus inhibiting caspase-dependent apoptosis [44]. This chaperone protein is responsible for the direct and indirect inhibition of both executioner caspases 3 and 7, mainly through ubiquitination of cytochrome c and procaspases [45,46,47,48]. Nevertheless, in our experiments p-HSP27 was significantly downregulated in the oviducts after 8R, suggesting an impaired anti-apoptotic function. Surprisingly, the protein content level of cleaved caspase 7 underwent a 3-fold increase in the treated group, whilst the cleaved caspase 3 remained unchanged. Due to the close interaction of oxidative stress and inflammation caused by the ovulation process, we can hypothesize a preferential activation of caspase 7 mechanisms, because of its pivotal role during cell proliferation and inflammatory response after the formation of inflammasomes [49,50,51]. The canonical apoptotic pathways require an extrinsic or intrinsic stimulus that activates the initiator caspases, leading to the cleavage of the executioner caspases for the cell death mechanism activation. Nevertheless, literature data demonstrate that the initial signal from the initiator caspases can be suppressed by the action of a family of proteins called inhibitor of apoptosis proteins (IAPs), among which we can find the X-linked inhibitor of apoptosis protein (XIAP) [52]. On this basis, to explain the different modulation of cleaved caspase 3 found in our experimental settings, we might hypothesize that it is a consequence of the presence of XIAP, in particular its baculovirus IAP repeat (BIR2) domain, which exerts its action efficiently on caspase 3 but not on caspase 7 [53,54]. These findings, together with the unaffected Bax protein content, suggest that, even though repeated cycles of hyperstimulation induce small but significant damage in mtDNA fragmentation and activate the antioxidant machinery, they do not trigger apoptotic mechanisms in oviductal cells.

The effect triggered by 8R of hyperstimulation induces oviductal cells to activate a pro-survival signaling pathway, initiated through the phosphorylation of p-p38 MAPK and p-MAPKAPK2, as reported elsewhere [55]. Indeed, both the high levels of p-p38 MAPK and the increased phosphorylation of SAPK/JNK, together with the known effect of p-AKT activation [23], promote the phosphorylation of c-Jun after 8R of gonadotropin stimulation. Noteworthily, the high p-c-Jun protein content agrees with our previous results, describing a statistically significant increase in cyclin D1 [23], a predominant downstream target of the c-Jun transcription factor [56]. Altogether, this panel of active kinases allows us to assume the establishment of a pro-survival signaling that could play a protective role against cell death and be in favor of tissue preservation. This hypothesis is strengthened by literature data on microglia BV-2 cells, showing the formation of a similar pro-survival and anti-apoptotic environment in the presence of p-SAPK/JNK and p-p38 MAPK kinases [57].

In conclusion, our study demonstrates that repetitive rounds of gonadotropin stimulation induce oxidative stress, as evidenced by the high protein content of catalase, GPx1, SOD-1 and -2 antioxidant enzymes, concomitantly with the modulation of pro-inflammatory apoptotic proteins p-HSP27 and caspase 7. Nevertheless, the oviducts engage the activation of a pro-survival kinase signaling pathway, driven by p-p38 MAPK, p-MAPKAPK2, p-SAPK/JNK and p-c-Jun, as a response to the exogenous stress caused by the repeated rounds of hyperstimulation. These findings, in combination with the previously published morphological and molecular data, augment the evidence suggesting that repeated rounds of ovarian stimulation affect the molecular landscape of oviducts, highlighting the imbalance of several biological processes. Thus, a high pro-inflammatory and oxidative status could further increase the risk of future tumor onset and development. This conclusion is supported by the recent finding that premenopausal and postmenopausal conditions contribute to increased cancer cell seeding only in the presence of an inflammatory reaction [58].

## 4. Materials and Methods

### 4.1. Chemicals

The chemicals used in this study were purchased from the following sources: rabbit monoclonal catalase (MAB-94599), SOD-1 (MAB-94600), SOD-2 (MAB-94601), Cleaved-Caspase-3 (D175) (AB-84283), mouse monoclonal GPx1 (MAB-94602) and ECL substrate (Pierce ECL-2001) from Immunological Sciences. Rabbit monoclonal Phospho-HSP27 (Ser82) (#9709), Phospho-c-Jun (Ser73) (#3270), Phospho-MAPKAPK-2 (Thr334) (#3007), Phospho-p38 MAPK (Thr180/Tyr182) (#4511), Phospho-SAPK/JNK (Thr183/Tyr185) (#4668), secondary antibodies goat anti-rabbit IgG conjugated to horseradish peroxidase (HRP; #7074) and horse anti-mouse IgG conjugated to horseradish peroxidase (HRP; #7076) from Cell Signaling Technology (Beverly, MA, USA). Mouse monoclonal DNA polymerase γ (sc-390634), Cleaved-Caspase-7 (sc-56063), Bax (sc-7480) and rabbit polyclonal actin (sc-1616) from Santa Cruz Biotechnology (Santa Cruz, CA, USA). Exgene^TM^ Genomic DNA Micro (cat. GA-118-050) was obtained from Geneall Biotechnology Co. Ltd. (Seoul, Republic of Korea) and Mouse Real-Time PCR Mitochondrial DNA Damage Analysis Kit (cat. DD2M) was purchased from Detroit R&D. All of the other reagents were purchased from Sigma-Aldrich (St. Louis, MO, USA), and were of the purest analytical grade.

### 4.2. Collection of Animals and Oviducts 

*Mus musculus* Swiss CD1 adult female mice (2–3 months old; Harlan Italy, Udine, Italy; *n* = 18) were housed in the animal facility under controlled temperature (21 ± 1 °C) and light (12 h light/day) conditions, with free access to food and water. Animals in the early luteal phase of the estrous cycle were confirmed via examination of vaginal smears [59,60,61] and used as Ctr. Repeated cycles of ovarian stimulation were performed according to the protocol utilized in our previous studies [22,23]. Briefly, mice were intraperitoneally injected with 5 IU of pregnant mare serum gonadotropin (PMSG) (Folligon, Milano, Italy) and 48 h later with 5 IU of human chorionic gonadotropin (hCG) (Corulon, Milano, Italy). Eight rounds (8R) of stimulation were performed with intervals of 1 week between each. Oviducts were collected from Ctr normo-ovulatory mice (*n* = 9) and animals hyperstimulated for 8R (*n* = 9), snap-frozen and stored at −80 °C for further analysis.

All the experiments were carried out in conformity with national and international laws and policies (European Economic Community Council Directive 86/609, OJ 358, 1 12 December 1987; Italian Legislative Decree 116/92, Gazzetta Ufficiale della Repubblica Italiana n. 40, 18 February 1992; National Institutes of Health Guide for the Care and Use of Laboratory Animals, NIH publication no. 85–23, 1985). The project was approved by the Italian Ministry of Health and the Internal Committee of the University of L’Aquila (2017). All efforts were made to minimize suffering. The method of euthanasia consisted of an inhalant overdose of carbon dioxide (CO_2_, 10–30%), followed by cervical dislocation. All the biological material used in this study was archived material properly stored soon after collection.

### 4.3. Mitochondrial DNA Damage

Total DNA was extracted from oviducts of Ctr and 8R mice, with the GeneAll^®^ Exgene^TM^ DNA micro kit following the manufacturer’s instructions. Briefly, the DNA concentration in the extracts was determined using Thermo Fisher’s Qubit™4 fluorometer (ThermoFisher Scientific, Waltham, MA, USA). mtDNA damage was quantified using the Mouse Real-Time PCR Mitochondrial DNA Damage Analysis Kit (Detroit R&D), according to the manufacturer’s instructions. This kit allowed the determination of 8.2 kb mtDNA fragments in mouse via quantification of the replicated DNA with real-time PCR following qPCR analysis. A total of 20 μL of PCR amplification reaction mixture was used, which contained 10 μL of 2x PowerUp SYBR Green Master Mix (AppliedBiosystems, Waltham, MA, USA), 0.9 μL of each primer (5 μM), 7.1 μL ddH_2_O (nuclease-free) and 2.0 μL of DNA. Real-time PCR conditions were 50 °C for 2 min, initial denaturation at 95 °C for 10 min, (40 cycles) 95 °C for 15 s and 60 °C for 60 s. Postamplification melting curve analysis was performed to confirm whether the nonspecific amplification was generated. All samples including the standard were amplified in triplicate. The standard curve was generated using the threshold cycle (CT) value and DNA concentration (logarithmic scale) of each of the 8.2 kb standards. Using linear regression analysis, the DNA concentration of the sample was determined based on the CT values. A high-level product of 8.2 kb represents less damage to mtDNA. The R^2^ for each standard curve was ≥0.99. In each run, negative and positive controls and a standard curve were included.

### 4.4. Western Blotting

Lysates from the oviducts (20 μg/sample) of Ctr and 8R mice were separated via electrophoresis onto 10% or 12% gels under reducing conditions and transferred to nitrocellulose membranes (Hybond C Extra, Amersham). Membranes were incubated overnight at 4 °C with specific primary antibodies in 5% *w*/*v* non-fat dry milk or in 5% *w*/*v* BSA: anti-catalase (1:1000), anti-SOD-1 (1:1000), anti-SOD-2 (1:1000), anti-GPx1(1:1000), anti-cleaved caspase 3 (1:500), anti-cleaved caspase 7 (1:1000), anti-Bax (1:200), anti phospo-HSP27(1:500), anti phospho-c-JUN (1:500), anti phospho-MAPKAPK-2 (1:500), anti pospho-38 MAPK (1:1000) and anti phospho-SAP/JNK (1:1000). HRP-linked anti-rabbit IgG (1:1000) and HRP-linked anti-mouse IgG (1:1000) were used as secondary antibodies (2 h, room temperature); peroxidase activity was detected using a chemiluminescence reagent (ECL, Pierce). The nitrocellulose membranes were examined using the Alliance LD2-77WL imaging system (Uvitec, Cambridge, UK). Densitometric quantification was performed with the public domain software NIH image v.1.62 and standardized using actin (1:2000) as a loading control.

### 4.5. Statistical Analysis

All experiments were repeated at least 3 times, and data are expressed as the mean ± SD. Student’s *t*-test was used for comparison between the two experimental groups (Ctr and 8R). Results were considered statistically significant when *p* < 0.05. All statistical analyses were performed using the statistical package SigmaPlot v.11.0 (Systat Software Inc., San Jose, CA, USA).

## Figures and Tables

**Figure 1 ijms-24-09294-f001:**
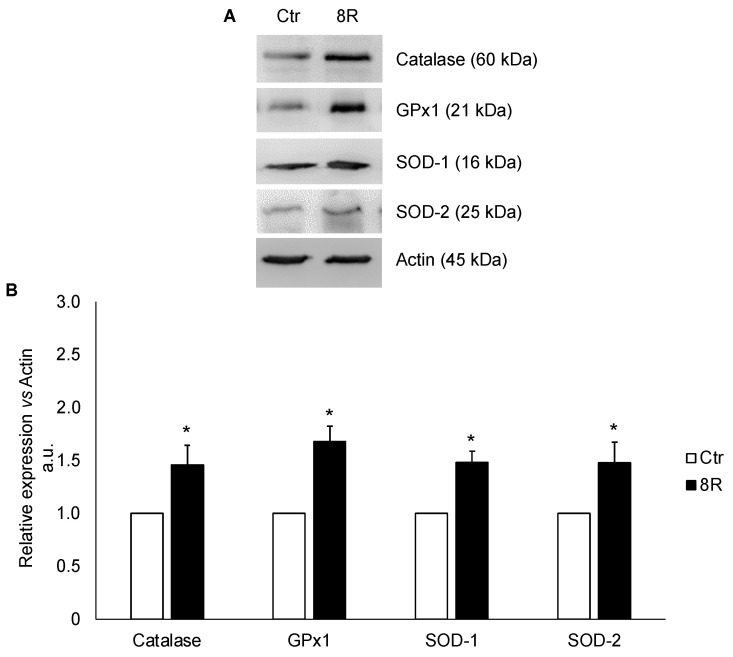
Antioxidant enzymes in oviducts collected from unstimulated (Ctr) and hyperstimulated (8R) mice. Representative Western blot images of catalase, GPx1, SOD-1 and -2 (**A**). Values are expressed as arbitrary units (a.u.), considering the Ctr values arbitrarily as 1. Bar graph data represent the mean ± SD after normalization of each protein with the respective actin used as a loading control for three independent determinations (**B**). * *p* < 0.05 vs. Ctr.

**Figure 2 ijms-24-09294-f002:**
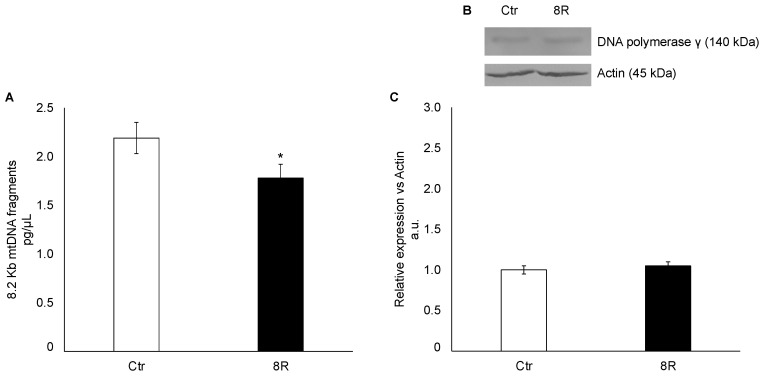
Mitochondrial DNA damage (**A**) and DNA polymerase γ protein content (**B**,**C**) in oviducts collected from unstimulated (Ctr) and hyperstimulated (8R) mice. Data are expressed as pg/µL of 8.2 Kb mtDNA fragments extracted from Ctr and 8R oviducts (**A**). Representative Western blot images of DNA polymerase γ (**B**). Protein values are expressed as arbitrary units (a.u.), considering the Ctr values arbitrarily as 1. Bar graph data represent the mean ± SD after normalization of each protein with the respective actin used as a loading control of three independent determinations (**C**). The results are presented as mean ± SD. * *p* < 0.05 vs. Ctr.

**Figure 3 ijms-24-09294-f003:**
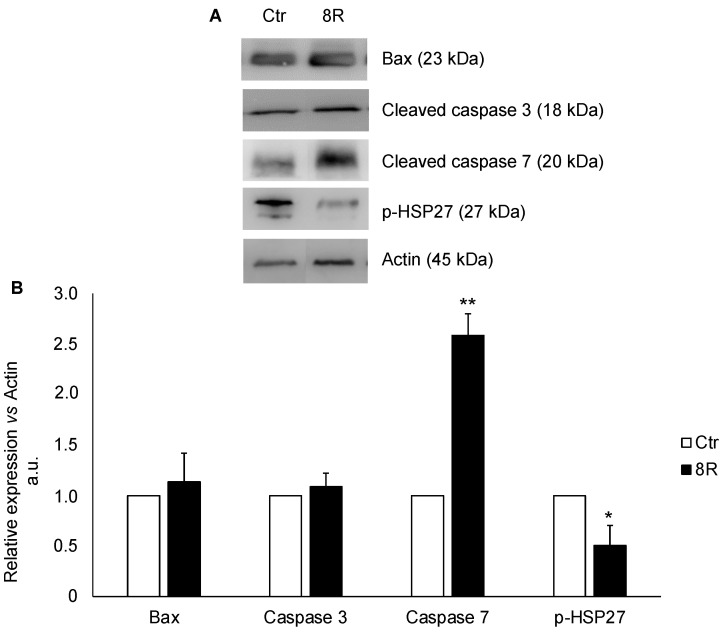
Apoptotic protein content in oviducts collected from unstimulated (Ctr) and hyperstimulated (8R) mice. Representative Western blot images of Bax, cleaved caspases 3 and 7 and p-HSP27 (**A**). Values are expressed as arbitrary units (a.u.), considering the Ctr values arbitrarily as 1. Bar graph data represent the mean ± SD after normalization of each protein with the respective actin used as a loading control of three independent determinations (**B**). * *p* < 0.05, ** *p* < 0.005 vs. Ctr.

**Figure 4 ijms-24-09294-f004:**
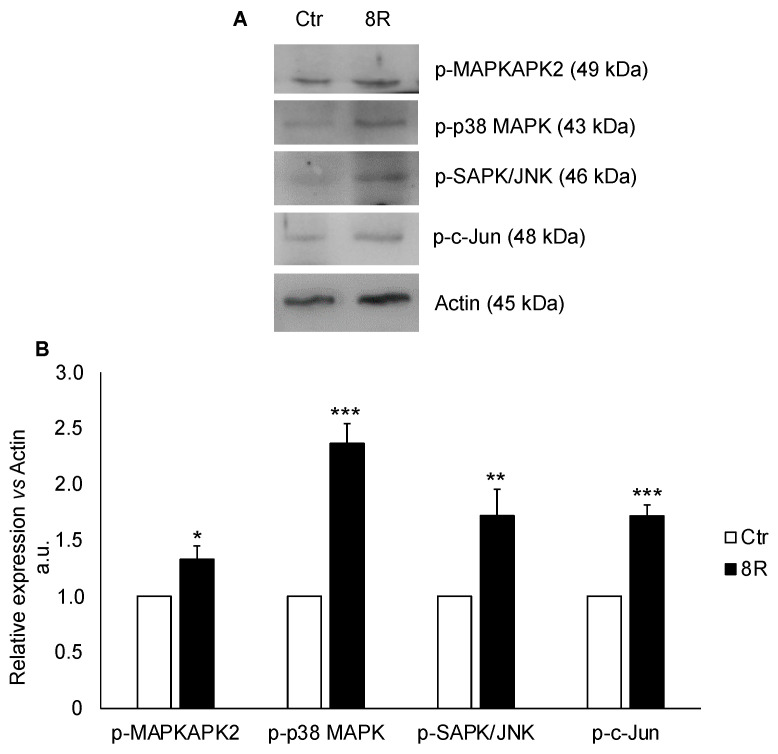
Cell-cycle-related protein content in oviducts collected from unstimulated (Ctr) and hyperstimulated (8R) mice. Representative Western blot images of phosphorylated (p)-MAPKAPK2, p-p38 MAPK, p-SAPK/JNK and p-c-Jun (**A**). Values are expressed as arbitrary units (a.u.), considering the Ctr values arbitrarily as 1. Bar graph data represent the mean ± SD after normalization of each protein with the respective actin used as a loading control of three independent determinations (**B**). * *p* < 0.05, ** *p* < 0.01, *** *p* < 0.001 vs. Ctr.

## Data Availability

The data presented in this study are available in the present article and on request to the corresponding author.

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
