# Peer review of "Repeated Rounds of Gonadotropin Stimulation Induce Imbalance in the Antioxidant Machinery and Activation of Pro-Survival Proteins in Mouse Oviducts"

_ijms, 2023, doi:10.3390/ijms24119294_

Round 1
Reviewer 1 Report
In this manuscript, authors report the effects of gonadotropin hyperstimulation on antioxidant enzymes, mitochondrial DNA damage, apoptosis-related proteins, and cell cycle regulators. Although the manuscript is interesting, a critical evaluation of the findings and conclusions is suggested. A few comments and suggestions are included below to improve the manuscript's quality.
The authors talk about inflammation and oxidative stress throughout the manuscript. Have authors previously demonstrated that 8R is inducing a pro-inflammatory environment? If not, I would recommend the authors include evaluations of the inflammatory status to demonstrate the “pro-inflammatory state” they mention after hyperstimulation.
In the abstract, the introduction is related to “inflammatory response” and the conclusion says, “the activation of pro-inflammatory mechanisms”. However, the title and evaluations are related to oxidative stress. Overall, the abstract is focused on inflammation rather than on the evaluations performed. Although oxidative stress conditions and inflammation are related, no direct inflammation evaluations were performed.
No clear hypothesis is stated in the abstract. Authors might want to make clear what the hypothesis is.
Line 19. Define ART. If “ART” is used only 2 times, maybe no abbreviation is needed.
Line 40. Authors state “the most used are generally based on”, the most used technologies? can authors modify the sentence to make it clear what they are talking about?
Lines 48-49. Reference is needed in “or reproductive cancers, i.e., breast, ovarian and endometrial cancer”
Lines 49- 51. Reference is needed in “ovarian cancer (OC) is a hormone related cancer, as oral contraceptive use and parity are well-established protective factors”
In the opinion of this reviewer the term “expression” (used here for protein levels here) refers to the expression of genes to make RNA, this reviewer’s suggestion is to use other terms instead to avoid confusion.
Line 85. Are those proteins exclusively “oxidative stress-related proapoptotic”? In the opinion of this reviewer, those proteins are general apoptotic proteins.
Line 96. The authors are using different formats throughout the paper to refer to 8 rounds of hyperstimulation: 8R, stimulated (8R).
Line 124. The authors name the R8 stimulated group as a “treated” group. This reviewer's suggestion is to homogenize how you are going to refer to both groups.
Line 111. The authors express the difference in the protein levels as a percentage but in lines 124-128 authors express the difference only using the p value. Authors might want to homogenize throughout the paper.
Fig 1. The 2 graphs have different formats; different graph size and space sizes between bars. One with the treatment group legend down, and the other on one side. Making all graphs as similar as possible is advised.
An explanation of B is missing in the figure description from Figure 1.
In Figure 2, the representative western blot for SOD-2 looks odd. Authors might want to make sure everything is fine.
Authors may want to homologue the “*P< 0.05 vs ctr” and “*P<0.05” in figures
Authors may want to homogenize the axis for graphs of protein relative levels.
Line 143. Is Figure 4, not Figure 3.
Lines 151-152. The authors state that at low levels ROS have functions in follicles. However, in line 166 authors state “…exposed to high levels of ROS present in follicular fluid…”. Can authors clarify or reorder ideas to make clear how ROS levels are in follicles?
Line 169. is “gamma” word appropriate in the context?
In the paragraph starting in line 183, the authors mention that “The overexpression of antioxidant enzymes…” prompted them to evaluate mitochondrial damage. However, in the results section, the figures follow a different rationale (mitochondrial damage is Fig 1). Can authors clarify or modify as needed? This reviewer’s suggestion is to have the same order in results and discussion as possible.
Lines 207-208. Authors state “During oxidative stress, p-HSP27 promotes cell protection during inflammatory response”, authors might want to rephrase and make sure if that is happening in oxidative stress, inflammation, or both processes based on the reference provided.
This reviewer strongly encourages the authors to address the evaluations of the pathways involved in differential activation of caspase 3 and 7 discussed in lines 211-224 to allow a better comprehension
Lines 222-224. The authors are redundant using “hyperstimulation” and “hyper stimulated” in the same sentence.
Lines 222-224. The authors conclude that hyperstimulation does not trigger apoptotic mechanisms. Can authors include a sentence about what is hyperstimulation doing so far?
Lines 240-241. The authors state “p-HSP27”, did the authors evaluate protein levels of p-HSP27 or of HSP27?
This reviewer strongly encourages authors to include evaluations of oxidative stress damage to support their observation of oxidative stress induction. Lipid oxidation, protein oxidation, or ROS could be evaluated.
In supplemental fig 3 authors show the western blots for HSP27. Can authors consider repeating the experiment or give a rationale for the second control showing a really low signal compared to the other controls?
It does not seem like P-p-38 representative western blots do not match with the ones in supplemental Fig 4.
Reviewer 2 Report
There were no major concerns in the manuscript. I just pointed out several comments which may improve the quality of the manuscript.
Can you explain why only 8 rounds of stimulation were used in the current study?
Have you considered the effect of intraperitoneal injection on oxidative stress in mouse oviducts? Why lost intraperitoneal injection control group?
Line 19, page 1. Please show the full name of ART as it is the first appearance in the text.
Line 143, page 5, Figure 3 should be Figure 4.
P > 0.05 and/or P < 0.05 should be P > 0.05 or P < 0.05 (italics), please check them through the whole manuscript.
Reviewer 3 Report
The manuscript by Valentina Di Nisio et al. shows that repeated rounds of gonadotropin stimulation induce oxidative stress in mouse oviducts. With the short of logics and supporting data, causal relationships is still not clear among : controlled ovarian stimulation (COS) through gonadotropin administration and oxidative stress. Several concerns about the quality of the data presented in the figures and figure legend , which require careful consideration.
Round 2
Reviewer 1 Report
The manuscript quality has improved. Below are some minor suggestions for the authors.
Line 90. Authors repeated the word “The”
Line 148-151. The sentence structure has the word “while” twice.
Figure 3. “B” missing from the updated Figure 3
In Fig 4 authors have “(B)” twice. Delete one
There might be a number of grammatical errors that need to be addressed. It might be helpful to have the manuscript read over by a “fresh pair” of eyes to catch some errors.
Reviewer 3 Report
I appreciate that the authors have done work in response to reviewers' concerns. However, this has not resulted in a manuscript with a clearer biological relevance. Instead, there are still some qualifications and limitations in the study, it is difficult to decipher what is established. Causal relationship is still not clear. Several concerns about the overall quality of the presented data remains unsolved enough. I can't support publication at IJMS.
